# Concurrent Validity of GAITRite and the 10-m Walk Test to Measure Gait Speed in Adults with Chronic Ankle Instability

**DOI:** 10.3390/healthcare10081499

**Published:** 2022-08-09

**Authors:** Ho Kim, Dongmin Kum, Insu Lee, Jongduk Choi

**Affiliations:** 1Department of Physical Therapy, Graduate School of Health and Medicine, Daejeon University, Daejeon 34520, Korea; 2Department of Physical Therapy, College of Health & Medical Science Daejeon University, Daejeon 34520, Korea

**Keywords:** chronic ankle instability, locomotion, gait speed, validation studies

## Abstract

Since there are many different assessments related to gait speed, it is important to determine the concurrent validity of each measure so that they can be used interchangeably. Our study aimed to investigate the concurrent validity of gait speed measured by the 10 m walk test (10 MWT) and the gold standard gait analysis system, the GAITRite system, for people with chronic ankle instability (CAI). For 16 people with CAI, 4 evaluations of the 10 MWT and 4 evaluations of the GAITRite system were performed (a comfortable gait speed for 2 evaluations; a maximal gait speed for 2 evaluations). We used intraclass correlations [ICC (2,1), absolute agreement] and Bland–Altman plots to analyze the relationship between the gait speed of the two measures. The absolute agreement between the 10 MWT and the GAITRite system is at the comfortable gait speed [ICC = 0.66; *p* < 0.001)], and the maximal gait speed [ICC = 0.68; *p* < 0.001)] showed fair to good agreement. Both gait speeds had a proportional bias; the limit of agreement (LOA) was large (0.50 at the comfortable gait speed and 0.60 at the maximal gait speed). Regression-based Bland–Altman plots were created for the comfortable gait speed (R^2^ = 0.54, *p* < 0.001) and the maximal gait speed (R^2^ = 0.78, *p* < 0.001). The regression-based LOA ranged from 0.45 to 0.66 m/s for the comfortable gait speed and 1.09 to 1.37 m/s for the maximal gait speed. Our study suggests that it is undesirable to mix the 10 MWT and the GAITRite system gait speed measurements in people with CAI. Each measure should not be recorded by the same evaluation tool and referenced to normative data.

## 1. Introduction

Walking speed is an effective, suitable, and reliable indicator for evaluating and monitoring overall health and functional status in various human situations. This sensitive ability has been designated as the”6th vital sign” [1]. Various disorders such as stroke, cerebral palsy, Parkinson’s disease, and chronic ankle instability (CAI) use walking speed to evaluate the condition, prescribe intervention, and track function [2,3,4,5]. Determining the concurrent validity of each measure is important in order to compare or interchangeably use the various assessments associated with gait speed [6].

A well-known, gold standard for gait analysis is the GAITRite system, a portable gait analysis tool for automatically measuring gait parameters. Many studies report that the GAITRite system is a reliable and valid evaluation tool [7,8]. Clinicians also conduct gait tests that can be performed easily without special equipment. The most representative test is the 10-m walk test (10 MWT) [1]. Recently, studies verifying the concurrent validity of the 10 MWT and the GAITRite system in patients with stroke, spinal cord injury, and healthy adults have been diversified [6,8,9,10].

People with CAI have an increased gait pattern of inversion kinematics and kinetics. Problems arise in that hip flexion increases in the terminal swing to midstance; hip extension decreases in the terminal stance to the initial swing; knee flexion increases; and the body weight shifts slowly at the beginning and end of the stance [11]. A previous study found no significant difference in the comparison of ankle kinematics and gait characteristics between subjects with CAI and healthy adults [12]. On the other hand, another study reported significant differences in the comparison of deviations in gait metrics between subjects with CAI and healthy adults [13]. As such, different variables of gait have different effects on subjects with CAI. However, no studies have evaluated the concurrent validity of gait speed measured by the GAITRite system and the 10 MWT in CAI with gait problems.

Our study aimed to evaluate the concurrent validity of gait speed in various conditions, measured with the GAITRite system and the 10 MWT, in people with CAI to determine whether the two assessments can be equally used.

## 2. Materials and Methods

### 2.1. Participants

Forty-one adults with suspected CAI who were admitted to Daejeon University as students in Daejeon, the Republic of Korea, were recruited. The participants’ selection criteria were as follows: (1) no gait disorder; (2) no neurological disorder; (3) a score of less than 24 on the Cumberland Ankle Instability Tool (CAIT); (4) a history of at least two episodes of ‘giving way’; and (5) feelings of ankle joint instability in the previously injured ankle joint. The criteria for excluding participants were as follows: (1) those with ankle pain; (2) those with a history of surgery on the lower extremities within 12 months; and (3) those diagnosed with a severe ankle sprain within 12 months [3]. Of the 41 recruited adults, 25 participants met the exclusion criteria and were excluded. Finally, 16 adults with CAI were selected for this study. Before the start of the study, the study procedure was explained to all participants, and written consent was obtained.

The experimental procedures for this study were reviewed by the Institutional Review Board and registered on the WHO International Clinical Trials Registry Platform: KCT0007155.

### 2.2. Study Design and Process

Our study has a cross-sectional design. The subjects performed a 10 MWT walk for 4 evaluations, and the GAITRite system (CIR systems Inc., Franklin, NJ, USA) walk for 4 evaluations. With both measurement tools, 2 of the walks were performed at a normal speed where the subject felt comfortable, and 2 walks were performed at the maximum speed at which the subject felt safe. In the case of the 10 MWT, marks were recorded at the beginning and end, so that the subject could recognize them, and measured using a stopwatch. The time elapsed from crossing the start mark to passing the end mark was manually recorded for each walk. The gait speed was calculated from the overall average time. The gait of the GAITRite system walk was calculated by dividing the elapsed time between the first and last steps. In both the 10 MWT and the GAITRite system walks, there was an acceleration and deceleration section of about 2 m. All subjects wore their usual running shoes [6]. The same tester walked with the subject to determine whether the subject had walked from the starting line to the finish line correctly (Figure 1).

### 2.3. Outcome Measures

#### 2.3.1. Cumberland Ankle Instability Tool

The Cumberland Ankle Instability Tool was the first questionnaire to score ankle instability. The questionnaire consists of nine questions with a total possible score of 30 points; a score of <24 is classified as ankle instability, and a score of 28 or more is classified as ankle stability. In the intra-rater reliability study of this questionnaire, ICC = 0.96, indicating high reliability and validity [14].

#### 2.3.2. 10 m Walk Test

As a method of evaluating walking speed while walking a straight distance of 10 m, a walking path was constructed in which the straight distance between two points 14 m apart was connected using tape. After setting the distance for acceleration and deceleration at 2 m at the beginning and end, the walking time was measured for a distance of 10 m of the walking path and used as a measurement variable for functional walking. This evaluation tool showed high reliability and validity [15].

#### 2.3.3. The GAITRite^®^ System

For the subject’s gait analysis, the GAITRite system (CIR systems Inc, Franklin, NJ, USA) equipment was used. This equipment is an electronic walking board with a length of 366 cm and a width of 61 cm. Six sensor pads are embedded inside the mat, and a special sensor is attached to the sidewalk. On the sensor pad, 2304 sensors in a 48 × 48 grid pattern are placed 1.27 cm apart. When the subject walks, the load of the subject’s feet is collected at 80 Hz per second, and this information is sent to the computer through a serial interface cable. The reliability of the evaluation items is 0.82–0.92, which is a high level [7].

### 2.4. Data Analysis

For the statistical processing of the subjects’ measured values, statistical analyses were performed using Windows SPSS ver. 25.0 (IBM Co, Armonk, NY, USA) and MedCalc ver. 19.3.1 (MedCalc software; Ostend, Belgium) programs. The absolute agreement between the gait speed of the 10 MWT and the GAITRite system was analyzed with an intraclass correlation coefficient (ICC 2,1). ICC was interpreted as poor (<0.40), fair to good (0.40–0.75), or excellent (>0.75) [16]. Linear Bland–Altman plots were constructed to evaluate the relationship between the 10 MWT and the GAITRite system. The limit of agreement (LOA) was calculated as mean difference values ± 1.96 * SD, and the difference values were calculated as 10 MWT speed—GAITRite speed. Regression-based Bland–Altman plots were constructed. Linear regression determined the mean proportional bias. Regression-based LOA was determined by the linear regression line ± SD. The significance level was set to α = 0.05.

## 3. Results

### 3.1. General Characteristics of the Subjects

The general characteristics of the subjects, including age, gender, height, weight, CAIT score, the 10 MWT gait speeds, and the GAITRite system gait speeds, are presented in Table 1.

### 3.2. Concurrent Validity

The absolute agreement between the 10 MWT and the GAITRite system is the comfortable gait speed [ICC = 0.66 (95% CI: 0.06, 0.88; *p* < 0.001)]; the maximal gait speed [ICC = 0.68 (95% CI: 0.11, 0.88; *p* < 0.001)] shows fair to good agreement.

The linear Bland–Altman plots for the comfortable gait speed [*p* < 0.001 (95% CI: 0.06, 0.19)] and the maximal gait speed [*p* < 0.001 (95% CI: 0.16, 0.33)] are shown in Figure 2A,C. LOA showed that the relationship between the measured values varied, with the comfortable gait speed of 0.50 and the maximal gait speed of 0.60. Regression-based Bland–Altman plots are shown in Figure 2B,D, with the comfortable gait speed (R^2^ = 0.54, *p* < 0.001) and the maximal gait speed (R^2^ = 0.78, *p* < 0.001). The regression-based LOA ranged from 0.45–0.66 m/s for the comfortable gait speed and 1.09–1.37 m/s for the maximal gait speed.

## 4. Discussion

Gait analysis is frequently used in clinical decision-making [17]. Therefore, it is essential to firmly validate the effectiveness of any gait analysis system prior to use in a clinical setting. The most commonly measured gait variables are speed and stride length, and the most commonly used method to evaluate speed is the 10 MWT [14]. Our study was aimed at adults with CAI by measuring the 10 MWT gait speed and the gait speed of the GAITRite system, a gait analysis system used to verify reliability and validity.

As a result, at a comfortable gait speed [ICC = 0.66 (95% CI: 0.06, 0.88) and fast gait speed [ICC = 0.68 (95% CI: 0.11, 0.88)], fair to good agreement between the 10 MWT and the GAITRite system was shown. These results showed the consistency of the two measurement methods for gait speed but, when absolute values of gait parameters are required (e.g., compared to normative values), they cannot be used interchangeably [18]. There was a systematic bias in the comfortable gait speed, where the gait speed of the 10 MWT was much faster than the GAITRite system. Both gait speeds had a proportional bias, and the LOA was large (0.50 at the comfortable gait speed and 0.60 at the maximal gait speed). A study measuring the concurrent validity of the 10 MWT and the GAITRite system in patients with chronic stroke found that both showed a large LOA (an LOA of 0.43 at the comfortable gait speed and an LOA of 0.37 at the maximal gait speed) [6]. Overall, the lack of agreement between these measures indicates that the 10 MWT and the GAITRite system do not demonstrate concurrent validity; therefore, the gait speed measured using these methods should not be used interchangeably. In this study, the walking speed of the 10 MWT was generally faster than the GAITRite system, which may reflect the difference in distance applied between each measurement.

In the case of the 10 MWT, more than 10 m was measured. However, in the case of the GAITRite system, the distance between the first and last measurements was short (3~4 m), so the measured walking speed of the 10 MWT may be more valid [6]. Unlike this study, the previous study confirmed that the walking speed of the GAITRite system was faster than a 3 m walk test (3 MWT). However, since the length of the GAITRite system walkway is 4.42 m, which is longer than the distance of the 3 MWT, it was reported that a higher speed might be achieved by accelerating over a longer distance [9], indicating that gait speed measurement over long distances could reduce the ratio of time-limited distances, including acceleration or deceleration. This study suggests that researchers should be careful when comparing the gait speed of the 10 MWT and the GAITRite system in people with CAI, and that it may not be appropriate to use them interchangeably. This approach means that information collected with the same gait speed measurement and evaluation tool should be used when measuring gait speed. The maximal gait speed of the concurrent validity may be slightly higher through the maximum level of fixation based on stabilization and by contracting the body muscles for fast movement. The comfortable gait speed may vary depending on the environment and the subject’s condition. These findings show that the results are consistent with studies on patients with chronic stroke [6]. However, the results indicate that it may be effective to measure maximal speed when comparing the gait speed of the 10 MWT and the GAITRite system; however, the concurrent validity is not the same in the end. This study is the first to examine the concurrent validity of the 10 MWT gait speed and the gold standard measuring tool, the GAITRite system, both of which are commonly used in clinical practice and target people with CAI. Peter [8] reports that it is necessary to investigate the inter-trial repeatability of spatiotemporal parameters in various pathologies to determine the effect of various gait patterns on the GAITRite system measurements’ reliability. We do not doubt that these studies will provide more diversity.

The limitations of this study are as follows: Firstly, the concurrent validity assessment is limited to determining agreement between measures, and it is not known which measures show values closer to the actual values. Secondly, although normality is established in this study, the small number of parameters was unfortunate. In future research, adding more personnel is necessary for conducting the experiment. Youdas [19] found that the difference in walking speed measured with a stopwatch and the GAITRite system was between −0.05 and 0.09 m/s. Therefore, since our study did not consider these measurement variables, they must be considered in future studies. Thirdly, the measurement of the GAITRite system and the 10 MWT could not be performed simultaneously. Since the length of the GAITRite system walkway used in this study was only 4.42 m, the measurement of the GAITRite system and the 10 MWT had to be carried out separately.

## 5. Conclusions

Our study suggests that it is undesirable to mix the 10 MWT and the GAITRite system gait speed measurements in people with CAI. Each measure should not be assessed with the same evaluation tool and referenced to normative data. We expect that various studies will be conducted on physical and pathological problems with various effects on gait parameters.

## Figures and Tables

**Figure 1 healthcare-10-01499-f001:**
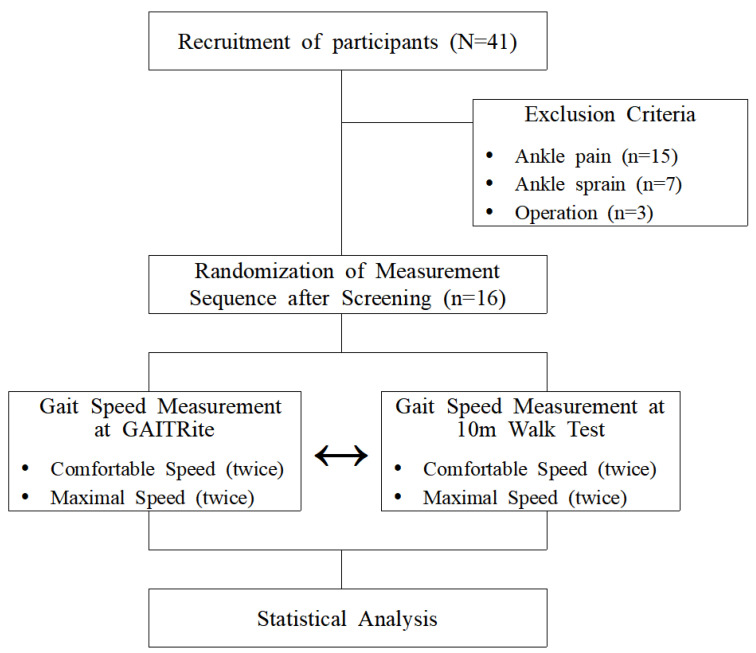
Flow chart.

**Figure 2 healthcare-10-01499-f002:**
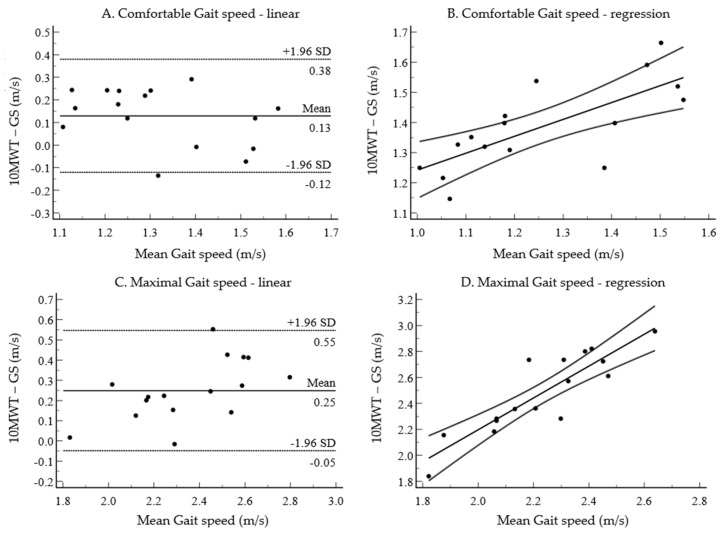
Bland–Altman plots. Relationship between mean gait speed of 10 MWT and GAITRite (GS) and difference between comfortable gait speed measurements (**A**,**B**) and maximal gait speed measurements (**C**,**D**). Mean difference values of (**A**,**C**) (center line) and linear LOA (upper and lower lines) are shown. The number to the right of each line is the y value. Lines (**B**,**D**) show the regression-based line of best fit (center line) and regression-based LOA (upper and lower lines) calculated from the standard deviation of residuals.

**Table 1 healthcare-10-01499-t001:** Demographics and gait speed of measurement tools.

Demographics	
Age (years)	23.62 (2.96)
Sex (Male/Female)	12/4
Height (cm)	173.18 (8.26)
Weight (kg)	72.93 (18.17)
CAIT (score)	15.74 (3.12)
Walking speed	
10 m walk test gait speed (m/s)	
*Comfortable*	1.38 (0.14); Range: 1.15, 1.66
*Maximal*	2.48 (0.30); Range: 1.84. 2.95
GAITRite System gait speed (m/s)	
*Comfortable*	1.25 (0.18); Range: 1.01, 1.55
*Maximal*	2.23 (0.22); Range: 1.82, 2.64

Values are expressed as Mean (Standard Deviation). CAIT, Cumberland ankle instability tool.

## Data Availability

The data presented in this study are available on request from the corresponding author. The data are not publicly available due to data privacy regulations.

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
