# Peer review of "Concurrent Validity of GAITRite and the 10-m Walk Test to Measure Gait Speed in Adults with Chronic Ankle Instability"

_healthcare, 2022, doi:10.3390/healthcare10081499_

Round 1

Reviewer 1 Report

Since there are many different assessments related to gait speed, it is important to determine the concurrent validity of each measure so that they can be used interchangeably.

The authors aimed to investigate the concurrent validity of gait speed measured by the 10 m walk test (10MWT) and the gold standard gait analysis system GAITRite (GS) for people with chronic ankle instability (CAI).

For 17 people with CAI, 4 evaluations of 10MWT and 4 evaluations of GS were performed (comfortable gait speed for 2 chances, maximal gait speed for 2 chances).

They performed a fine statistical analysis.

The authors concluded that: (a) it is undesirable to mix 10MWT and GS gait speed measurements in people with CAI. (b) Each measure should be recorded by the same evaluation tool and referenced to normative data.

The study is interesting and faces an attractive topic.

These are my minor comments.

1.     Better explicit the aim. Now it seems described very synthetically with the text “Our study aimed to evaluate the concurrent validity of walking speed as measured by GS and 10MWT in people with CAI.

2.     I’d avoid to use the acronym GS for GAITRite and simply I' d like to refer to GAITRite

3.     Introduction needs to be enlarged for a scientific article, citing for example other reference systems and the importance of these tests, different for example from the tests on a tapis-roulant.

4.     A photo could enrich the presentation.

5.     A flow chart could be useful to improve the readability of the design.

6.     Check the resolution of the figures.

Author Response

Summary of Revision

The authors deeply appreciate the voluntary contribution of each reviewer in the form of valuable comments that helped to improve our manuscript. Every effort has been taken by the authors to revise the manuscript according to the reviewers’ comments.

<Reviewer 1> Comments:

  1. Better explicit the aim. Now it seems described very synthetically with the text “Our study aimed to evaluate the concurrent validity of walking speed as measured by GS and 10MWT in people with CAI.
  • We greatly appreciate your comment, and agree with your opinion. As per your recommendation, we have revised the sentences. The aim of this study has been revised to be more clear.

  1. I’d avoid to use the acronym GS for GAITRite and simply I' d like to refer to GAITRite
  • Modified according to the reviewer's request.

  1. Introduction needs to be enlarged for a scientific article, citing for example other reference systems and the importance of these tests, different for example from the tests on a tapis-roulant.
  • We greatly appreciate your comment, and we agree with your opinion. We have added and revised additional rationale for conducting our study. Thank you for pointing it out.

  1. A photo could enrich the presentation.
  • Unfortunately, there are no prepared photos, so it is impossible to present them in the thesis. I also think that the figure already presented can replace the photo.

  1. A flow chart could be useful to improve the readability of the design.
  • We greatly appreciate your comment, and we agree with your opinion. We have inserted a flowchart.

  1. Check the resolution of the figures.
  • All authors involved in our study really appreciate your kindness and suggestions. We modified according to the reviewer's request.

Reviewer 2 Report

Comments to the Authors

General Comments:

The authors of this manuscript sought to evaluate the utility of the GAITRite system in individuals with CAI. Overall, I believe that the authors did a fine job from a study design perspective; however there are several methodological concerns I have and would need to be clarified before acceptance. Additionally, several portions of the manuscript are hard to follow and the readability needs to be improved.

Introduction:

Line 30. “Various diseases…” Consider rephrasing as stroke and CAI are not diseases.

Line 36. “Many studies report…” only one citation is given. Either add the additional citations or rephrase. Several of these additional studies I believe would be beneficial in other aspects of this manuscript as well.

The introduction needs to include aspects of how CAI impacts gait velocity as that is the variable being assessed. Balasukumaran et al 2020 showed no differences between controls and CAI with gait velocity. Gigi et al 2015 found differences in gait velocity in control and CAI groups.

Besides the population, being used what makes this a novel study or advances the literature forward? The need for this study is not clear to me at this point.

As the overall GS system has been shown to be a valid tool in other populations why would that not transfer to this population?

Methods:

Why were the GS trials and the 10MWT trials collected separately? If you are looking to examine if the velocity is the same with the two devices it should be done across the same trials. The other studies investigating the validity of the GS collected the criterion method at the same time as the GS data. This is a major concern of this study.

Additionally, with the use of a stopwatch over a 10 meter test is questionable. Did the tester walk along side the participant to accurately assess when the crossed the start and finish line? Was the same tester used for all 10MWT? Providing the reliability data of the 10MWT between trials would be of great benefit.

Discussion:

I feel that the discussion fits the data that is presented adequately, however a more detail discussion of the limitations of both the tests I believe beneficial and when one should be used over the other. I feel that may help in addressing the novelty concern from the introduction and can add to a more robust discussion of the application of the findings.

Author Response

Summary of Revision

The authors deeply appreciate the voluntary contribution of each reviewer in the form of valuable comments that helped to improve our manuscript. Every effort has been taken by the authors to revise the manuscript according to the reviewers’ comments.

<Reviewer 2>  Comments:

  1. Line 30. “Various diseases…” Consider rephrasing as stroke and CAI are not diseases.
  • All authors involved in our study really appreciate your kindness, suggestions and correction of wording errors. After collecting comments from reviewers, corrections were made.

  1. Line 36. “Many studies report…” only one citation is given. Either add the additional citations or rephrase. Several of these additional studies I believe would be beneficial in other aspects of this manuscript as well.
  • We greatly appreciate your comment, and agree with your opinion. As per your recommendation, we have revised the sentences. Previous studies have been added and described.

  1. The introduction needs to include aspects of how CAI impacts gait velocity as that is the variable being assessed. Balasukumaran et al 2020 showed no differences between controls and CAI with gait velocity. Gigi et al 2015 found differences in gait velocity in control and CAI groups.
  • We are very grateful for your recommendation, and agree with your opinion. As per your recommendation, we have spent a significant amount of time in improving the introduction section in the revised manuscript. Thanks to the reviewers, the quality of this study has been improved, thank you.
  1. Besides the population, being used what makes this a novel study or advances the literature forward? The need for this study is not clear to me at this point.
  • We greatly appreciate your comment, and agree with your opinion. As per your recommendation, we have revised the sentences. The aim of this study has been revised to be more clear. CAI affects various variables of gait. Therefore, while there are various evaluations of the elements of gait, this study was conducted to find out whether it can be used equally by examining the concurrent validity of GAITRite, which is the gold standard, and 10mwt, which is widely used in clinical practice.

  1. As the overall GS system has been shown to be a valid tool in other populations why would that not transfer to this population?
  • First of all, I think that there is a big difference in the characteristics (disease, disability et al) of the subject to be measured. In addition, Cleland (2019) studied the concurrent validity of GAITRite and 10MWT in a study on stroke patients, showing high validity only at maximal speed.

  1. Methods: Why were the GS trials and the 10MWT trials collected separately? If you are looking to examine if the velocity is the same with the two devices it should be done across the same trials. The other studies investigating the validity of the GS collected the criterion method at the same time as the GS data. This is a major concern of this study.
  • This was added to the limitations of this study. Thank you so much for the reviewer's advice.

  1. Additionally, with the use of a stopwatch over a 10 meter test is questionable. Did the tester walk along side the participant to accurately assess when the crossed the start and finish line? Was the same tester used for all 10MWT? Providing the reliability data of the 10MWT between trials would be of great benefit.
  • Thanks for the reviewer's advice, we have accepted this part and made the correction.

Round 2

Reviewer 2 Report

The manuscript has improved and overall comments were addressed satisfactorily.